

# High resistance to climatic variability in a dominant tundra shrub species

Victoria T. González[1,2], Mikel Moriana-Armendariz[1], Snorre B. Hagen[2], Bente Lindgård[1], Rigmor Reiersen[1] and Kari Anne Bråthen[1]

[1] Institute of Arctic and Marine biology, University of Tromsø, Tromsø, Norway
[2] Department of Ecosystems in the Barents region, Norwegian institute of Bioeconomy Research-NIBIO, Svanvik, Norway

## ABSTRACT

Climate change is modifying temperature and precipitation regimes across all seasons in northern ecosystems. Summer temperatures are higher, growing seasons extend into spring and fall and snow cover conditions are more variable during winter. The resistance of dominant tundra species to these season-specific changes, with each season potentially having contrasting effects on their growth and survival, can determine the future of tundra plant communities under climate change. In our study, we evaluated the effects of several spring/summer and winter climatic variables (i.e., summer temperature, growing season length, growing degree days, and number of winter freezing days) on the resistance of the dwarf shrub *Empetrum nigrum*. We measured over six years the ability of *E. nigrum* to keep a stable shoot growth, berry production, and vegetative cover in five *E. nigrum* dominated tundra heathlands, in a total of 144 plots covering a 200-km gradient from oceanic to continental climate. Overall, *E. nigrum* displayed high resistance to climatic variation along the gradient, with positive growth and reproductive output during all years and sites. Climatic conditions varied sharply among sites, especially during the winter months, finding that exposure to freezing temperatures during winter was correlated with reduced shoot length and berry production. These negative effects however, could be compensated if the following growing season was warm and long. Our study demonstrates that *E. nigrum* is a species resistant to fluctuating climatic conditions during the growing season and winter months in both oceanic and continental areas. Overall, *E. nigrum* appeared frost hardy and its resistance was determined by interactions among different season-specific climatic conditions with contrasting effects.

## INTRODUCTION

Climate change is taking place across all seasons. In tundra ecosystems in particular, there is evidence of a long term warming trend during winter, spring and summer (*Epstein et al., 2017*) resulting in longer growing seasons and varying snow cover conditions (*Xu et al., 2013*). An increase in extreme winter warming events, where snow melts in midwinter and exposes tundra vegetation to subsequent frost damage, has also been reported (*Bjerke et al., 2017*). Studying how dominant tundra species respond to this season-specific

Corresponding author
Victoria T. González,
victoria.gonzalez@nibio.no

climatic variability, can help predict the potential resilience of tundra communities under climate change.

Species with traits providing reduced sensitivity to environmental variability, such as long-life span and seed bank accumulation (i.e., storage effects) (*Chesson, 2000*), or with plastic growth rates (*Jump & Penuelas, 2005*), are expected to have high overall resistance to climate change (*Oliver et al., 2015*). Here we define resistance as the capacity of a species to remain stable in the face of environmental perturbations through persistence (i.e., positive growth and reproduction) (*Oliver et al., 2015*; *Ingrisch & Bahn, 2018*). Thus, under the predicted climate warming scenario, a resistant species should grow and reproduce at a similar or higher rate under warmer conditions. For example, shrubs, which account for much of the biomass in tundra ecosystems (*Walker et al., 2005*), appear resistant to increasing summer temperatures, responding with increased biomass and flowering (*Myers-Smith et al., 2011*; *Buizer et al., 2012*). However, some shrubs display poor resistance signs, such as shoot death and low productivity, in response to changing winter conditions, e.g., extreme winter warming events (*Bjerke et al., 2014*). Hence, the overall resistance of tundra shrubs to climatic variability appears to be season-dependent and is currently not well understood.

A common approach for studying the impact of climate change on plant resistance is through manipulation experiments during the growing season (e.g., *Elmendorf et al., 2012*). However, experimental studies have been found to under predict the effects of climate change (*Wolkovich et al., 2012*). Further, there are calls for the integration of summer and winter conditions in climate studies, especially in seasonally snow-covered ecosystems (*Sanders-DeMott & Templer, 2017*). Thus, observational studies encompassing a range of ongoing yet differing climate change developments during all seasons, can provide insight in to the long-term adaptive potential of species and help predict their resistance to climate change.

*Empetrum nigrum* is an evergreen dwarf shrub that creates vast areas of monospecific vegetation in tundra ecosystems across the northern hemisphere (*Tybirk et al., 2000*). *E. nigrum* is a niche constructing species and its ecosystem modifying properties are well documented (*Wardle et al., 1998*; *Tybirk et al., 2000*). For example, *E. nigrum* has been linked to a reduction in species richness in tundra plant communities (*Mod et al., 2016*; *Bråthen, Gonzalez & Yoccoz, 2018*), and has been found to slow down the recovery of tundra heathland after simulated winter warming damage mainly due to its allelopathic properties (*Aerts, 2010*). Further, *E. nigrum* has responded to warmer growing seasons with an increase in biomass, flowering, and fruiting in both manipulation experiments (*Buizer et al., 2012*) and observational studies (*Bråthen, Gonzalez & Yoccoz, 2018*). However, there are also reports of decreasing reproductive and vegetative output caused by winter extreme events (*Bokhorst et al., 2010*; *Bokhorst et al., 2011*) and outbreaks of the pathogenic fungus *Arwidssonia empetri* under increased snow cover (*Olofsson et al., 2011*). Nevertheless, *E. nigrum* thrives in a wide range of habitats, from exposed ridges to more sheltered depressions and appears to have high morphological plasticity to cope with varying snow cover conditions (*Bienau et al., 2014*), in addition to being resistant to ice encapsulation (*Preece, Callaghan & Phoenix, 2012*). Hence, the future abundance and distribution of

*E. nigrum* in tundra ecosystems under climate change is currently not well understood. Since *E. nigrum* has a strong influence on plant community structure and diversity, hence reducing plant community resilience (*Oliver et al., 2015*), understanding its response to climate change is imperative for making predictions of the vast areas where it is present.

The goal of our study was to investigate the resistance of *E. nigrum* to fluctuations of temperature and precipitation over multiple winter and summer seasons along a steep climatic gradient. To achieve this, we measured over six years shoot growth, berry production, and vegetative cover in five *E. nigrum* dominated tundra heathlands, in a total of 144 plots covering a 200-km gradient from oceanic to continental climate. Using soil-surface temperature loggers, we registered how often the vegetation was exposed to freezing temperatures (i.e., number of freezing days) in addition to fluctuations in growing season length, summer temperature and growing degree days. We expect *E. nigrum* to be a species capable of persisting under varying environmental conditions because it has a broad ecological niche occurring along a gradient of temperature, moisture and bedrock types (*Norwegian Biodiversity Information Center (NBIC), 2005*; *Bråthen, Gonzalez & Yoccoz, 2018*), it is able to dominate heathlands under a range of climates (*Büntgen et al., 2015*), and is a slow growing, long lived species (*Bell & Tallis, 1973*). However, we still expect *E. nigrum* fitness to be defined by climatic conditions across all seasons and hypothesize that shoot length and berry production will be dependent on an interaction of growing season and winter conditions. Further, we aim to confirm (a) that higher summer temperatures and longer growing seasons can give longer shoots and more berries (e.g., *Buizer et al., 2012*) and, (b) that freezing exposure (e.g., *Bokhorst et al., 2009*) or shallow snow cover (*Bienau et al., 2014*) can give shorter shoots or decrease *E. nigrum* biomass and cover.

## MATERIAL AND METHODS

### Study sites

The study took place between 2010 and 2016 in five *E. nigrum* dominated tundra heathlands in northern Norway between 69.17–70.02°N and 18.75–20.9°E (Fig. 1). The study sites were located at the *Betula pubescens* tree line ecotone along a 200-km climatic gradient, ranging from oceanic to continental conditions. The sites were chosen due to their contrasting climatic conditions, that is, coastal sites are normally characterized by cool summers, mild winters and varying snow cover conditions, while continental sites have warm summers and extremely cold winters which allow for a stable and deep snow cover. Snow cover at the study sites usually remains until mid-June and bedrock is mainly gabbro (Table 1). Besides *E. nigrum*, other less abundant species across all sites were *Betula nana*, *Vaccinium uliginosum*, *Vaccinium myrtillus* and *Vaccinium vitis-idaea* (*PanarcticFlora, 2017*).

### Study design and sampling

In June 2011, ten blocks were established in each of the sites (Fig. 1). Blocks had at least 90% *E. nigrum* cover, a maximum slope of 5 degrees and were placed a minimum of five meters apart to avoid including the same *E. nigrum* individual in different blocks, since long-lived, clonally reproducing species can have multiple modular units (*Miller, 2012*).

**Figure 1  Study sites.** (A) Location of study sites. The names and Continentality Index of the sites from coast to inland were: (1) Rebbenes (21.8), (2) Skogsfjord (23), (3) Snarby (24.2), (3) Skibotn (30.1), and (4) Gukhesjávri (31); and (B) Study design.

**Table 1  Study site characteristics.** Environmental variables were registered on site (i.e., elevation, soil pH and soil organic horizon), and climatic temperature variables were calculated using on-site temperature loggers placed at soil surface level during all six study years. Precipitation data, snow depth and bedrock were gathered from publicly available databases (i.e., http://www.senorge.no).

|  | Rebbenes | Skogsfjord | Snarby | Skibotn | Guhkesjávri |
|---|---|---|---|---|---|
| Latitude/Longitude | 70°10′; 18°45′ | 69°57′; 19°15′ | 69°45′; 19°30′ | 69°14′; 20°33′ | 69°10′; 20°42′ |
| Continentality index | 21.8 | 23 | 24.2 | 30.1 | 31 |
| Mean annual temp. (°C) | 2.2 | 0.5 | 0.6 | −2 | −1.9 |
| Mean summer temp. (°C) | 12.37 | 11.77 | 12.04 | 11.96 | 11.64 |
| Elevation (m a.s.l.) | 10 | 264 | 266 | 580 | 510 |
| Continentality index | 21.8 | 23 | 24.2 | 30.1 | 31 |
| Soil pH | 4 | 4 | 4.5 | 3.5 | 4 |
| Soil organic horizon depth (cm) | 5–10 | 3–10 | 2–5 | 3–10 | 5–10 |
| Mean annual prec. (mm) | 871.18 | 860.49 | 917.19 | 634.16 | 628.2 |
| Mean summer prec. (mm) | 218.75 | 179.86 | 172.16 | 189.75 | 202.98 |
| Snow depth (cm/year) | 2.41 | 12.9 | 25.2 | 39.2 | 39.9 |
| Bedrock | Amphibolite | Basalt | Mica gneiss, schist and metasandstone | Metasandstone and schist | Gneiss and migmatite |

Each block consisted of three 50 cm × 50 cm permanently marked plots placed beside each other with a 40-cm separation. Two blocks were unfortunately dropped during the study due to diverse issues (one in Skibotn and one in Gukhesjávri, Fig. 1), hence we had a total of 48 blocks and 144 plots for vegetation analyses.

Each year, all sites were sampled within one or two days from each other at the end of the growing season in late August. *E. nigrum* cover and berry production were measured with the use of a wooden frame (50 cm × 50 cm) divided in 16 subplots (0.0125 m$^2$ each) by registering presence/absence of live *E. nigrum* biomass and *E. nigrum* berries in the 16 subplots. Shoot length was measured by selecting five different annual shoots each year (i.e., shoots grown during that summer) in each plot (total of 15 shoots per block). In 2017, *E. nigrum* biomass per plot was measured using the point intercept method with

20 pins per plot (*Bråthen & Hagberg, 2004*) and converting the total number of hits using established calibration functions (*Ravolainen et al., 2010*).

## Climatic variables

Temperature and precipitation based climatic variables were used in this study (Table 2). Three temperature loggers (Thermochron iButtons®) were placed at ground level at each site (i.e., one at the top, middle and bottom of study site) to measure soil surface temperature every 3 h all year round (during 2010–2016) and were used to calculate all temperature variables. Mean daily temperatures per site were calculated as the average of the three temperature loggers. Number of freezing days and annual climatic variables were calculated from sampling date to sampling date (i.e., from late August to late August next year) since *E. nigrum* flower and vegetative buds are fully formed at the end of the growing season and are therefore subjected to any climatic variability during the previous autumn/winter (*Bell & Tallis, 1973*). Hence, the number of freezing days for each sampling year belongs to the previous winter season. We calculated three growing degree day variables by summing all daily mean temperatures above 1 °C (GDD+1), 2 °C (GDD+2) and 5 °C (GDD+5) starting from the date when the average daily logger temperature was above 1 °C, 2 °C or 5 °C respectively, for three or more consecutive days until the sampling date. Although the common threshold in northern climates to determine both growing season and growing degree days is set to >5 °C (GDD+5) (*Körner, 1999*), *E. nigrum* is known to both flower and start growing straight after snow melt (i.e., GDD+1 or GDD+2), and we were interested in testing this assumption. We further calculated the number of winter warming extreme events (EE) during December-March, by summing the periods of more than two days when air temperatures were above +2 °C (*Bokhorst et al., 2012*) and soil surface temperatures were above +1 °C (suggesting absence of snow). Precipitation variables were gathered from publicly available maps and online databases (*seNorge, 2017*). The Continentality Index for each study site was calculated according to (*Rivas-Martinez, Rivas-Saenz & Penas, 2011*) (37), showing a range of 21.8 to 31 indicating a gradient from oceanic/maritime to continental climate.

## Statistical analyses

All data were sampled at plot level, averaged to block level and analyzed using the statistical environment R (*R Development Core Team, 2016*). An exploratory analyses of the data was done using non-metric multidimensional scaling (NMDS) set to two dimensions with a Bray-Curtis dissimilarity test in the "vegan" package in R (*Oksanen et al., 2018*). NDMS allowed us to visualize and interpret the relationship between the climatic variables and our response variables with the use of rank orders. The exploratory variables used in the NMDS included all climatic variables (Table 2) except the extreme event variable (EE), because we recorded no extreme events except three episodes on the island of Rebbenes. We also included biomass measured in 2017 in the NMDS analyses and we assumed that, since *E. nigrum* is a slow growing species, the biomass measured in 2017 was a good overall representative measure for each site.

Further, linear mixed effect models (*Pinheiro et al., 2017*) were used with block as the random factor and mean shoot length or mean berry frequency as response variables. The

**Table 2 Definition of environmental variables used in this study.** All temperature variables were calculated from on-site temperature loggers. Precipitation was gathered from the publicly available database http://www.senorge.no.

| Variable | Definition |
| --- | --- |
| Freezing days (FD) | Sum of days between sampling dates where the mean daily temperature was below $-1$ °C. Hence, the number of freezing days for each sampling year belongs to the previous winter season. |
| Summer temperature (ST) | Mean daily temperature of June, July and August (until sampling date). |
| Growing season length (GSL) | Sum of days from the date where the average daily logger temperature was above 5 °C for three or more consecutive days until sampling date. |
| Growing degree days (GDD+1, GDD+2, GDD+5) | Sum of all daily mean temperatures above 1 °C (GDD+1), 2 °C (GDD+2) and 5 °C (GDD+5) starting from the date when the average daily logger temperature was above 1 °C, 2 °C or 5 °C respectively, for three or more consecutive days until the sampling date. |
| Growing season precipitation (GSP) | Mean daily precipitation from start of the growing season until sampling date (i.e., same days as GSL). |
| Non-growing season precipitation (NGSP) | Mean daily precipitation from sampling date until start of the next growing season (i.e., until start of GDD+1). |
| Extreme events (EE) | Sum of periods of more than two days when air temperatures were above $+2$ ° C, and surface temperatures were above $+1$ ° C (suggesting absence of snow). |

choice of predictor variables was based on the exploratory NMDS results and Pearson correlation values between variables. We used Akaike Information Criterion (AIC) to rank the models and chose the final model with the lowest AIC value (*Johnson & Omland, 2004*).

# RESULTS

## Variability along the climatic gradient of biotic and abiotic variables

Both biotic and climatic variables varied between years and along the climatic gradient (Figs. 2 and 3). Mean shoot length varied among sites and was between 0.65 and 3.2 cm (mean of 1.3 cm across sites), with shorter shoots found in continental sites (Fig. 2A). Shoot length varied little between years (Fig. 2B). Mean berry frequency, in contrast, had a larger variation between years and among sites (Figs. 2C–2D). The cover of *E. nigrum* during the six years was constant in all plots showing 16 out of 16 subplots with *E. nigrum* live biomass (supplemental raw data). We did not register any large-scale mortality or browning during the study in any of our study plots (VT González, 2010–2016, pers. obs.). *E. nigrum* biomass registered in 2017 increased with continentality index, finding more biomass in continental areas, whereas mean shoot length was on average shorter in continental areas (Fig. 4).

There was a sharp contrast in temperature between study sites along the climatic gradient during the winter months (Fig. S1), finding a fluctuating snow cover (i.e., soil surface temperature loggers registered below freezing temperatures) mainly at the two
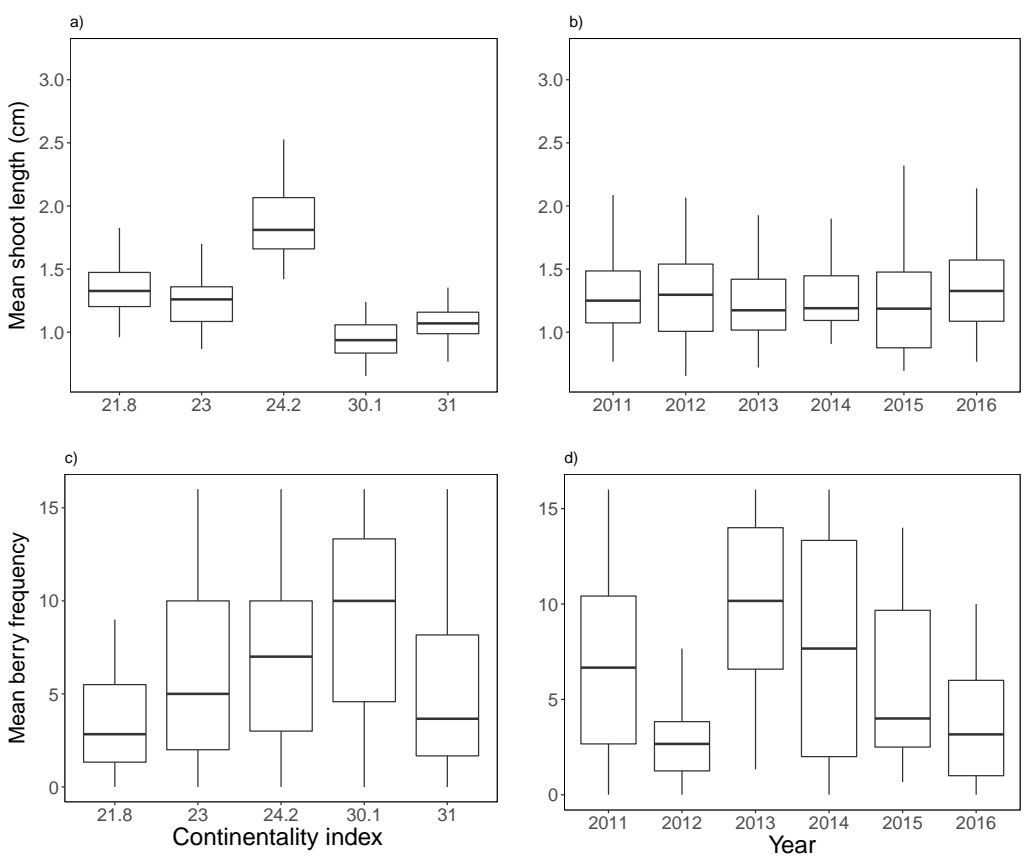

**Figure 2** **Biotic variable variation between years and along the climatic gradient.** Mean shoot length (A–B) and mean berry frequency (C–D). Boxplots are presented with median and non-overlapping boxes are indicative of statistical significance.

most continental sites (Fig. S1). Among the climatic variables, the number of freezing days (FD), non-growing season precipitation (NGSP) and growing season length (GSL) were contrasting between sites rather than between years, finding three times fewer freezing days and twice the precipitation at the coast (Figs. 3A, 3K, 3G). The two most continental sites had many freezing days each year, registering on average seven months a year with freezing temperatures (Fig. S1), though FD were registered along the entire climatic gradient and in all years (Fig. S1). Mean summer temperature (ST) varied less among sites and more between years (Fig. 3F), while growing degree days (GDD) and growing season precipitation (GSP) varied between both sites and years (Figs. 3C, 3D, 3I, 3J). No extreme winter warming episodes (EE) were registered except for the island of Rebbenes where three episodes occurred between 8–10 February 2012, between 4–5 December 2014, and finally between 15–18 of March in 2015.

## Exploratory analyses

The nonmetric multidimensional scaling (NMDS) exploratory analyses had a stress value of 0.09 indicating a good fit of the data with the number of selected dimensions (i.e.,

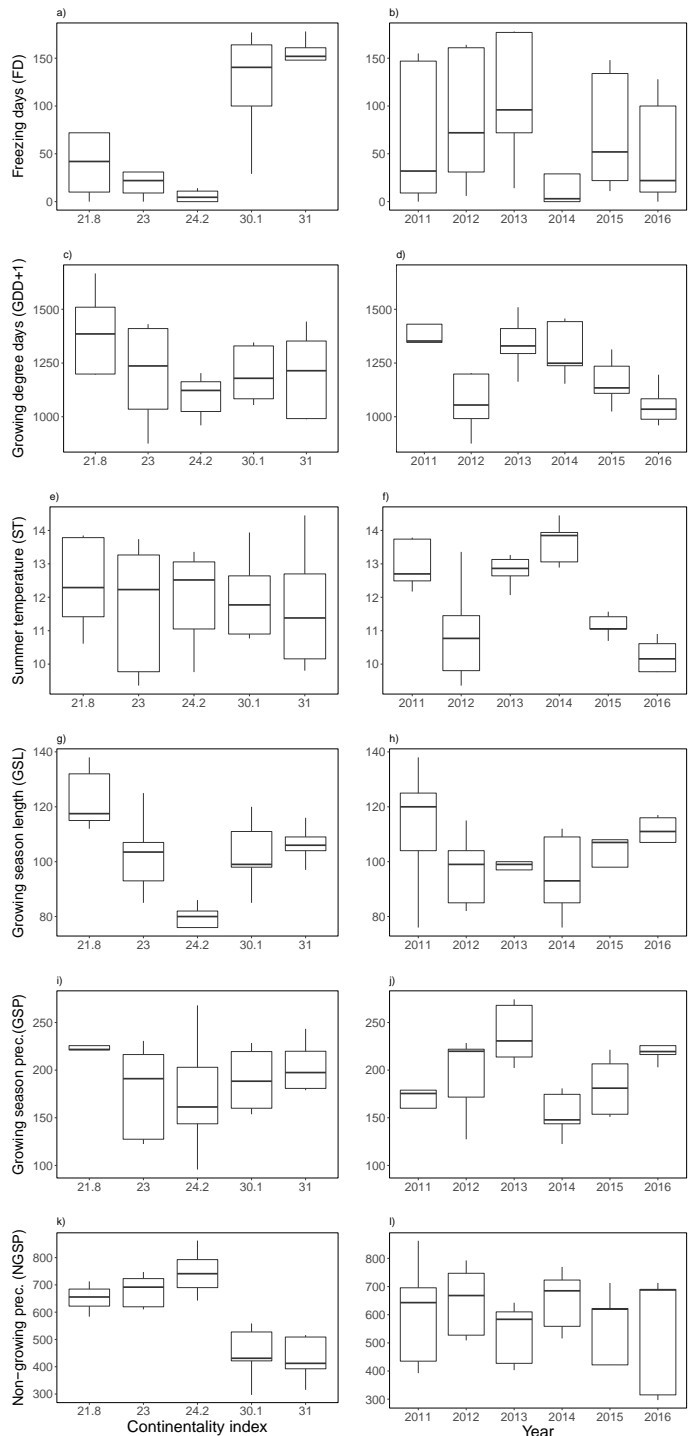

**Figure 3  Variation between years (i.e., 2011–2016) and along the climatic gradient (i.e., Continentality Index) of abiotic variables.** Freezing days (FD) (A–B), growing degree days (GDD+1) (C–D), summer temperature (ST) (E–F), growing season length (GSL) (G–H), growing season precipitation (GSP) (I–J) and non-growing season precipitation (NGSP) (K–L). Boxplots are presented with median and non-overlapping boxes are indicative of statistical significance.

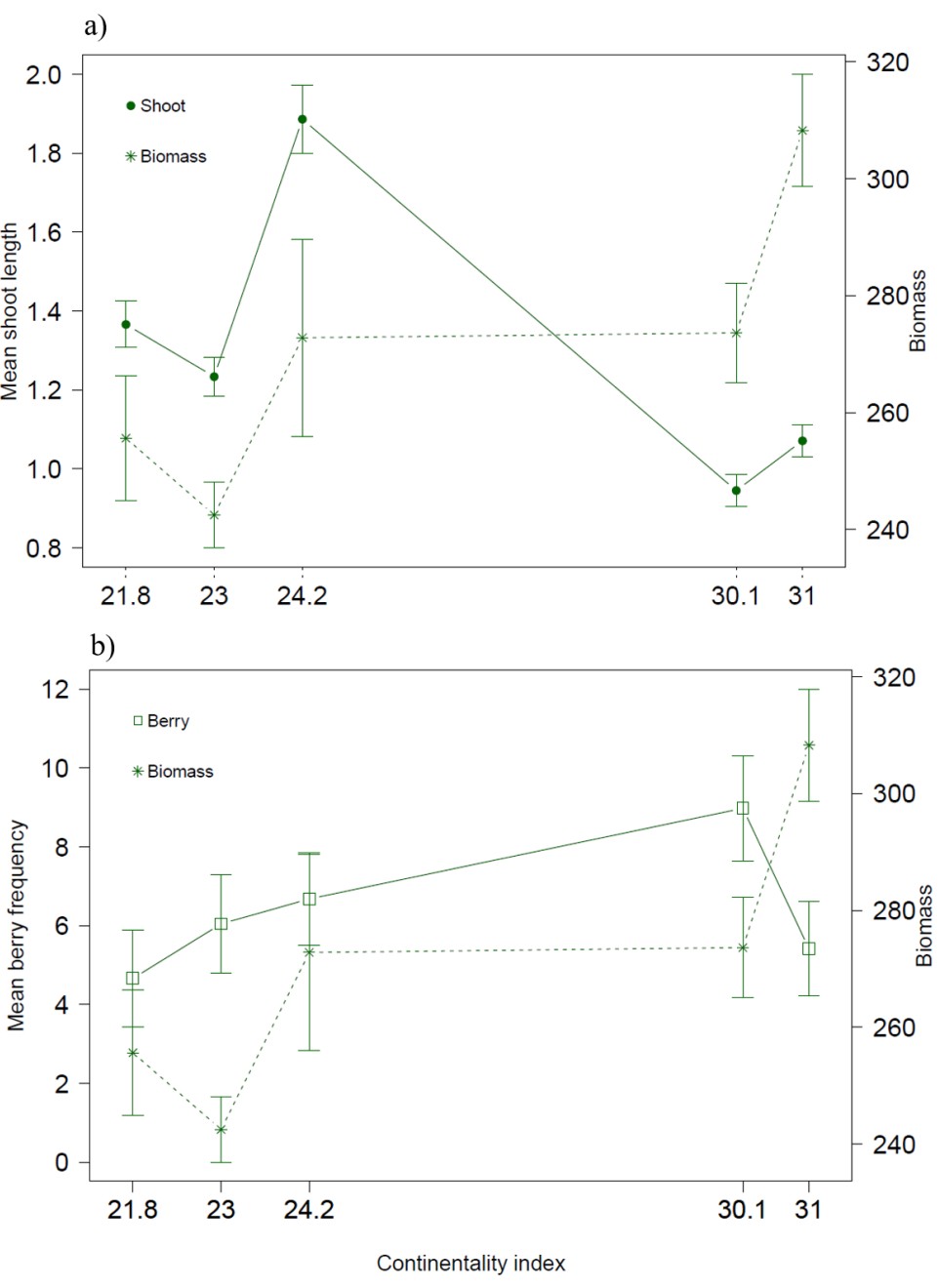

**Figure 4** **Biotic variables along the climatic gradient.** Biomass $(g/m^2)$ together with (A) mean shoot length (cm) and (B) mean berry frequency.

two). Mean shoot length was mainly related to NMDS axis 1 and partly to axis 2, whereas berry frequency was largely related to axis 2. Further, the response variables differed in their correlation with the ordination space $(R^2)$ finding that berry frequency had a low correlation value $(R^2 = 0.01)$ while mean shoot length was better correlated with the ordination space $(R^2 = 0.43)$. The climatic variables varied also in the relation with the NMDS axes. The

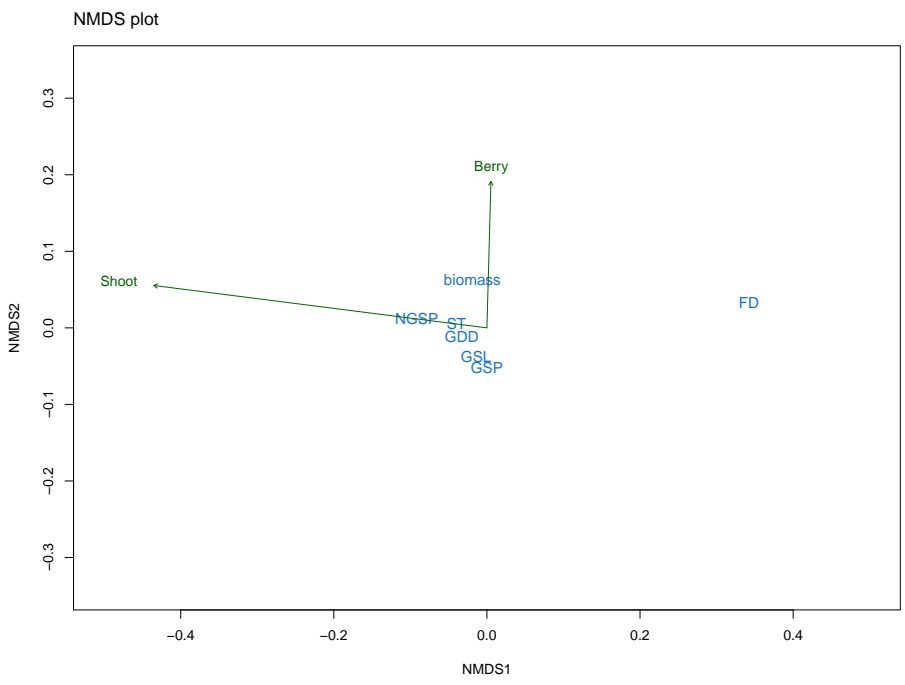

**Figure 5  NMDS plot from exploratory analyses.** The plot illustrates the relationship between response variables (green arrows): mean shoot length (Shoot) and mean berry frequency (Berry) and predictor variables (in blue) with the ordination space: freezing days (FD), growing degree days (GDD), summer temperature (ST), growing season length (GSL), growing season precipitation (GSP) and non-growing season precipitation (NGSP).

first axis was strongly related with FD, followed by GSP and NGSP (Fig. 5). The second axis was related with all other variables, that is, growing season length (GSL), summer temperature (ST), all three growing degree day variables (GDD+1, GDD+2, GDD+3) and biomass (Fig. 5). Overall, from the first NMDS axis, FD was the best correlated variable with the ordination space (Fig. 5) while from the second NMDS axis, although biomass appeared as the strongest variable, the effect of all variables was small and similar (Fig. 5). Further, all sampling years appeared to be climatically similar (Fig. S2A) and there was a climatic overlapping among sites, except for the two most continental sites (continentality index (CI) of 30.1 and 31) and the mid-continental site (CI of 24.2) (Fig. S2B).

The Pearson correlation tests among the predictor variables selected by the NMDS, showed that several variables were correlated to each other (Table S1). A main finding was that FD was correlated with all other variables except GDD (all three). Further, GDD (all three) were strongly positively correlated with ST and GSL (Table S1). Thus, to test the main hypothesis of our study (i.e., understanding the effect of climatic fluctuations during all seasons on *E. nigrum* resistance), the NMDS results and Pearson correlation values suggested the best climatic variables were the number of freezing days (FD) and the growing degree variables (GDD+1, GDD+2, GDD+5), which represented winter and spring/summer seasons respectively.

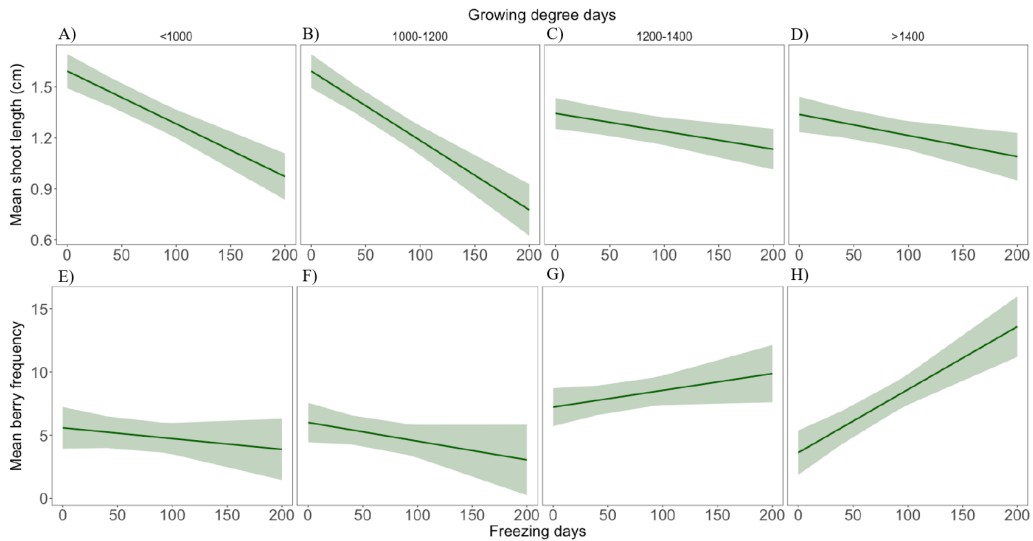

**Figure 6** **Interactive effect of number of freezing days and growing degree days on response variables.** Mean shoot length (A–D) and mean berry frequency (E–H). The mean is presented with 95% confidence intervals.

### *E. nigrum* resistance under climatic variability

We selected through the Akaike information criterion (AIC) GDD+1C, out of all three growing degree day variables, in interaction with FD as the optimal predictor-variable combination since it had the lowest AIC. We used this combination for both mean shoot length and mean berry frequency as response variables.

Results from the linear mixed effects models showed a significant interaction between FD and GDD+1 on both mean shoot length and mean berry production (Table S2 ). Thus, during colder growing seasons there was a negative effect of an increase of FD on mean shoot length and milder negative effect on berry production, while under warmer growing seasons this negative effect was less apparent in shoot length and turned positive in the case of berry production (Fig. 6).

## DISCUSSION

Our study illustrates how the common evergreen dwarf shrub *E. nigrum* is resistant to fluctuating climatic conditions during the growing season and winter months by showing positive vegetative growth and reproductive output in all sites and in all years. Exposure to freezing temperatures during winter was correlated with shorter shoot growth and reduced berry production the following season, however, if the following season was warm and long it appeared to compensate for these negative effects. Thus, *E. nigrum* fitness was affected by climatic conditions during both winter and spring/summer.

Exposure to extreme low temperatures can damage vegetation either directly through freezing or winter desiccation, or indirectly through ice encasement after rain-on-snow episodes and refreezing of melted snow (*Blume-Werry et al., 2016*; *Bjerke et al., 2017*). Under varying snow cover conditions, evergreen dwarf shrub foliage is likely subjected

to freezing temperatures, as their branches might protrude from the snowpack. In our study, *E. nigrum* appeared to be regularly exposed to freezing temperatures during large parts of the winter months but was nevertheless able to grow and reproduce even during colder growing seasons, which suggests that this species is generally frost hardy and, as experimental studies have shown, possibly resistant to ice encapsulation (*Preece, Callaghan & Phoenix, 2012*; *Preece & Phoenix, 2014*). In contrast to studies showing large mortality of *E. nigrum* in tundra heathlands after extreme winter warming events (*Bokhorst et al., 2009*; *Bjerke et al., 2014*), we did not find indications of browning episodes during the course of the study in spite of the low temperatures registered at most sites during the winter months. One possible explanation is that in contrast to winter warming episodes, we did not register above zero temperatures during the winter months (except for the three short episodes at the coastal site of Rebbenes) indicating spring-like development was not initiated and the plant remained dormant. *E. nigrum* experiences deep dormancy during the winter months and has been found able to remain undamaged by temperatures down to −40 C (*Körner, 1999*). Nevertheless, we did not see any indications of browning of the vegetation in our plots at the coastal site of Rebbenes the years it did experience winter warming episodes.

The mean shoot lengths registered during our study were similar to those registered in other *E. nigrum* studies (*Wipf, Rixen & Mulder, 2006*; *De Witte & Stöcklin, 2011*; *Bienau et al., 2014*). The small variation in shoot length between years confirms that *E. nigrum* is a slow growing species with a conservative growth strategy. Thus, our assumption that the biomass measured in 2017 was an appropriate overall representation of biomass amounts of each site was supported by this finding. It is worth noting that *E. nigrum* biomass was higher at continental sites, but the shoot length was the shortest. In shallow or varying snow cover conditions, *E. nigrum* has been found to have short internodes and to form low mats close to the ground as to avoid having apical growth stems constantly exposed to the freezing temperatures (*Bienau et al., 2014*). This in turn could be a selective force on growth during summer, that is, sites with low snow cover might have densely structured dwarf shrubs with a larger number of shoots, as indicated by biomass amounts, but shorter as to be less exposed during winter months. Thus, winter conditions could affect *E. nigrum* structure and hence modify the growth pattern during the summer months, reflecting the complex interactions between all seasons on this common tundra species.

We found a negative effect of increasing freezing days on mean shoot length, however, warmer growing season conditions appeared to compensate for this effect, confirming our hypothesis that the overall resistance of this species is defined by both spring/summer and winter conditions. The evergreen dwarf shrub *Cassiope tetragona*, has recently been found to respond to damage by experimental winter freezing conditions with enhanced shoot growth (*Milner et al., 2016*). Though we did not monitor yearly shoot mortality or damage, our study suggests that *E. nigrum* might respond in a similar manner to winter freezing exposure, by showing a plastic growth rate dependent also on spring and summer climatic conditions. Nevertheless, *E. nigrum* showed positive shoot growth in all sites during all years indicating the resistance of this species to climatic variability.

Berry production was also found dependent on spring/summer and winter conditions and appeared to be specially promoted by warmer and longer summers, which could

indicate that *E. nigrum* berry production could increase under climate warming. However, we did also find a slight increase of berry frequency with continentality that also could be due to an increase in biomass, as berry frequency was mostly related with NMDS axis 2 which corresponded with biomass amounts. This would confirm, in line with previous studies, increasing amounts of biomass to be associated with higher flowering and fruiting (*Buizer et al., 2012*; *Kaarlejarvi et al., 2012*; *Bråthen, Gonzalez & Yoccoz, 2018*). Thus, the increase in berry production associated with increasing freezing days and warmer summers could be a direct result of an increase in biomass due to compensatory shoot growth stimulated by freezing during the colder seasons. *E. nigrum* flowering buds are formed during the previous autumn and both flower and berry production have been found unaffected by experimental winter warming treatments and icing episodes (*Preece, Callaghan & Phoenix, 2012*). Thus, a higher reproductive output might be expected if the growing season conditions are optimal (long and warm growing seasons as showed in this study) despite the colder winter conditions. Nevertheless, freezing or icing-induced enhancement of vegetative or reproductive output has important implications for climate change studies and as to avoid biased conclusions both winter and growing season conditions should be considered.

It is worth noting that growing season length starting with +1 degrees Celsius explained most variation in both shoot length and berry productivity. Some evergreen species are known to break winter dormancy even before snow melt, as soon as light penetrates a shallow snow cover (*Körner, 1999*). It appears *E. nigrum* is able to start growth and reproduction very early during the growing season, which could be advantageous in light of the predicted earlier spring snowmelt following climate change (*Wipf, 2010*; *Krab, Roennefarth & Becher, 2018*). Hence, the predicted increase in growing season length with climate change could promote *E. nigrum* encroachment in areas where it is dominant. Recent studies have particularly showed an encroachment of *E. nigrum* in tundra areas (*Vowles et al., 2017*; *Vuorinen et al., 2017*; *Maliniemi et al., 2018*) which might be explained by its ability to respond to varying climatic conditions as shown here, and its niche construction ability (*Bråthen, Gonzalez & Yoccoz, 2018*).

## CONCLUSIONS

Overall, our study highlights the synergistic effect of all seasons on the growth and reproduction of the common evergreen dwarf shrub *E. nigrum*. The findings presented here, suggest that *E. nigrum* is frost hardy and able to persist under varying winter temperature conditions by showing increased reproductive and vegetative output under warmer growing season conditions, thus further perpetuating the positive feedback surrounding shrub expansion in the tundra in connection with climate change (*Myers-Smith et al., 2011*). Our study further supports the importance of better understanding the linkages between all seasons on the impact of climate change on dominant tundra species.

## ACKNOWLEDGEMENTS

We are thankful to Sissel Kaino, Xavier Ancin, Anna-Katharina Pilsbacher, Metha Klock and Mildrid Svoen for help during the field work.

### Funding

The publication charges for this article have been funded by a grant from the publication fund of UiT The Arctic University of Norway. The funders had no role in study design, data collection and analysis, decision to publish, or preparation of the manuscript.

### Grant Disclosures

The following grant information was disclosed by the authors:
UiT The Arctic University of Norway.

### Competing Interests

The authors declare there are no competing interests.

### Author Contributions

- Victoria T. González and Kari Anne Bråthen conceived and designed the experiments, performed the experiments, analyzed the data, contributed reagents/materials/analysis tools, prepared figures and/or tables, authored or reviewed drafts of the paper, approved the final draft.
- Mikel Moriana-Armendariz performed the experiments, analyzed the data, contributed reagents/materials/analysis tools, prepared figures and/or tables, approved the final draft.
- Snorre B. Hagen analyzed the data, contributed reagents/materials/analysis tools, prepared figures and/or tables, authored or reviewed drafts of the paper, approved the final draft.
- Bente Lindgård and Rigmor Reiersen conceived and designed the experiments, performed the experiments, contributed reagents/materials/analysis tools, approved the final draft.

### Data Availability

The raw measurements are available in the Supplemental File. The raw data shows all data collected in the field, both biological measurements (mean berry frequency and mean shoot length) and abiotic (climatic) measurements (for abbreviations used see main text).

### Supplemental Information

Supplemental information for this article can be found online at http://dx.doi.org/10.7717/peerj.6967#supplemental-information.

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
