# Peer review of "High resistance to climatic variability in a dominant tundra shrub species"

_PeerJ, doi:10.7717/peerj.6967_

## Round 0.1 · original submission · Minor Revisions

The reviewers have judged your paper as well worth publishing in PeerJ, and I agree. They each had some suggestions for further improvement that you can see below. Please address all the comments.

Reviewer 1 ·

Basic reporting

No comment.

Experimental design

No comment.

Validity of the findings

No comment.

Additional comments

This article adds much needed data to improve understanding on vegetation dynamics in the Arctic under different scenarios of climate change. It is particularly good that you have 6 years of data, and an impressive number of plots and sites. The article is well written with good, clear English throughout. I do not find any major problems with the work, but offer some advice for minor changes to help improve the manuscript.

Line 20. Not sure why it is important to say ‘allelopathic dwarf shrub’ here, as it does not seem relevant to anything else in the abstract. If you want to allude to its role as a ‘niche constructor’ perhaps try and phrase it slightly differently. But this is just a suggestion.

66-71. This section about the importance of Empetrum as niche constructor species is very interesting and highlights its importance in the ecosystem. However it would be useful to have a bit more detail about its positive and negative effects. e.g. When you say it ‘has been found to mediate the positive effects of increasing summer temperatures in plant community structure’ it would be useful to know how/by what mechanism. Similarly, how does it ‘reduce the recovery of tundra heathland’?

90. For the second hypothesis I find it a bit unnecessary to say ‘varying winter conditions’, as this is quite vague/unclear, and then immediately after you refer to freezing exposure and shallow snow. So I would recommend changing the sentence to something like ‘b) that freezing exposure (ref) and shallow snow cover (ref) can give shorter shoots…’.

108. Missing word. Should say ‘Empetrum thrives IN a wide range of habitats’.

111. Correct ‘in addition to be resistant’ to ‘in addition to being resistant’.

170. I think index needs a capital letter to start – Index.

190-199. Looking at figures 2,3 and 4, and reading the accompanying text in the results it is hard to know if there are any statistically significant differences between sites or years. I would make this clear in the text specifying significance if relevant, and/or in the figures with the use of asterisks or letters to denote differences.

205. Change ‘three times less’ to ‘three times fewer’.

227 and 229. Can you please specify Fig S2 a or b for extra clarity.

242-247. These results about the interaction between freezing days and GDD are very interesting and perhaps counter-intuitive, as it seems that with high GDD, there is a positive relationship between FD and berry production. I was hoping for some explanation of this finding in the discussion (such as in the section 304-311). As this is not simply that Empetrum is tolerant to freezing, but even that it has a benefit.
Please try to offer some account for why this might be, and possible implications this could have for berry production in light of possible changing winter conditions. Is it that the freezing stimulates some kind of compensatory growth, as discussed earlier in the manuscript for shoot growth? Would this therefore be able to be maintained for year after year, or Empetrum resources would run out, and berry numbers would fall again?

266. Change ‘what suggests’ to ‘which suggests’.
305 and 316. Change ‘what could’ to ‘which could’.
307. Change ‘what could’ to ‘that could’ (or with a comma before, it can be ‘continentality, which could’.)
318. Comma not need between change and could.

329. Be careful, there is a full stop instead of a comma after ‘Furthermore’. Or perhaps you don’t need this word.

Fig. 3 caption. Be careful that there is a space after ‘years’ and between ‘of abiotic’.

Fig 4. Again check there is a space between ‘the climatic’. At least on the review version it seems not to have it. Also, I would increase font size of axes titles (numbers seem ok) and legend.

Fig. 5. The font size of predictor variables is small, and at the moment the growing degree days and other variables are all written over each other, so it is hard to read. Maybe use points with labels offset, or don’t put all the GDD labels on, as it is not very readable as it is.

Fig. 6. Also seems to be some words without spaces between, but am now thinking this is a problem with the formatting of the review version.

Fig S2. I know its “only” supporting information but I would recommend increasing font size on FigS2. Also it would be good to add a more informative title to each plot, so it can be seen easily which is Continental Index and which is sampling year, especially as these seem to appear in the opposite order in the text (lines 227-229). Perhaps change this around.

Yours sincerely,
Catherine Preece

·

Basic reporting

The manuscript is written in scientifically appropriate English.

The data is presented in several figures, which mostly are appropriate. The description of what is shown in the graphs, however, could sometimes be improved. For instance, Figure 6 has eight panels, which are separated into two groups by berry/shoot growth. It is unclear from the figure label what the horizontal separation means, though, and even in the text I can roughly conclude what the categories must mean based on the interpretation, rather than come to the interpretation based on the meaning of the categories.

The literature cited is ample, though in some places better publications exist than the ones being cited, even though they may be older. A prime example of this is line 52: the two Myers-Smith papers cited here are a) a paper using satellite data and b) a paper studying shrubs across a treeline in Canada. There are papers discussing the effect of experimental warming on E. nigrum specifically, such as Parsons 1994 (J of Ecology 82(2) 307-318), which even does so in the Scandinavian subarctic.

The structure of the article is in the professional standard and appropriate.

The hypotheses are clearly stated, and are discussed in the Discussion.

Experimental design

The submitted article asks the question whether climate change will change the fitness and performance of Empetrum nigrum, a panarctic, ubiquitous dwarf shrub. Rather than the experimental approach that is often used to answer these questions, the authors use a gradient of continentality and year-on-year variation as a proxy for climate change. This is an important test to establish whether Empetrum nigrum, in the long run, will disappear or be outperformed by other shrub species as the arctic changes with changing climate.

Validity of the findings

I have a general issue with the use of “resistance” in this manuscript. The authors argue that E. nigrum is resistant to climate variation because there is always positive shoot growth. I would argue that there being positive shoot growth is a sign of E. nigrum being alive, rather than resistant to climate variation, and that in fact the correlation between GDD/freezing days and shoot growth suggests that E. nigrum does respond to climate change – if there was resistance, in my understanding there would not be a response to climate variation. This is not a flaw in the study per se and need not affect the manuscript beyond a simple rewording of the conclusion and relevant parts of the Discussion.

Additional comments

These are some specific comments based on the manuscript line numbers:

35: This is true, however, as you state in the next sentence, while warming occurs across all seasons, most warming has been shown to occur in winter and spring – I think this should be mentioned here (an old reference for this would be Serreze et al 2000, Climatic Change 46:159-207, although I’m sure more recent ones also exist).
58: I think more importantly, there are weaknesses to experimental warming/any ecosystem experiment that do not exist for a large-scale gradient such as the one you use here, so this is something worth mentioning here.
66 and ff: I am aware that colloquially, you would refer to E. nigrum as Empetrum, but in this case the correct scientific notation would be E. nigrum (or Empetrum sp if you would like to refer to the genus). This is an issue throughout the manuscript.
66: I am not entirely sure what this sentence means – how has E. nigrum mediated the positive effect of summer warming? You do not need to re-state your previous paper here, but this is a fairly unclear statement.
76: I think part of the problem is that even six years, which is an admirably long time for an experiment, is likely to be too little time to observe a complete die-back of one or more species, especially if you are measuring plants that, by definition, are there, rather than the ones that are not there. To a certain extent the nature of the study (the climate gradient) compensates for this, but here, too, you are basically stating that E.nigrum can still exist in the places where it was dominant six years ago, rather than that it necessarily maintains its competitive fitness.
86: Even if you define resistance as you do, the important question here surely will often not be whether a single species (in this case E. nigrum) can grow under varying climate conditions, but how the response which you are anticipating compares to that of other dominant species. Ie, will other shrubs outcompete Empetrum, which is slow growing as you state later in the article.
108: should read “E. nigrum thrives IN a wide range of habitats” I assume.
94-113: I think this would be better merged with the Introduction. It seems to give the rationale for working with this particular organism, so it would be better suited there.
124-129: I am unsure why this is relevant – surely if you are monitoring herbivory (large-scale herbivores, that is), the interesting part here would be if the grazing pressure differs between sites, rather than which herbivores are present? (I may be missing something)
142: was there a standard for which shoot was selected (eg, apical, second from apical, etc)?
179: It would be worthwhile restating which variables these are specifically here.
183: is there a reason why site was not included as a random variable here? Was it included in the model at all elsewhere?
239: this is a very minor quibble, but I think this should read “we selected GDD+1C based on the AIC”…
266: “which suggests that…”
256-258 and 267: Something about this juxtaposition seems off to me; the point about ice encasement is an important one (especially since your gradient study may verify experimental findings, although this may be hard to prove considering you could not look at ice formation on the plants – this is not a criticism of the study, there is only so much time!), but you spend ten lines explaining why ice encasement may affect vegetation, before citing two papers that say experimentally it does not affect Empetrum nigrum, which is the only species you are interested in. Maybe rewriting this paragraph to make the experimental finding clearer earlier may work better.
305: “which could indicate”
316: “which could be advantageous”
318: It could, but only if E. nigrum is the only species that responds this way to an increase in GDD – if other species respond in the same way or even show a more pronounced response, then the result could be that E. nigrum is outperformed. I think this consideration of comparative competitiveness should be mentioned somewhere in the manuscript.


Overall, I think this is a study containing a lot of hard work and some valuable data, so the above points are hopefully useful for a possible revision. The manuscript may need, in my opinion, some clarifications, but no new data analysis or data.

---

## Round 0.2 · accepted · Accept

The suggestions made by the reviewers were followed, improving the article that can now be published.

#